# From Backward Spreading to Forward Replay: Revisiting Target Construction in LLM Parameter Editing

**Wei Liu** [* 1]   **Hongkai Liu** [* 1]   **Zhiying Deng** [2]   **Yee Whye Teh** [3]   **Wee Sun Lee** [1]

## Abstract

LLM parameter editing methods commonly rely on computing an ideal target hidden-state at a target layer (referred as anchor point) and distributing the target vector to multiple preceding layers (commonly known as backward spreading) for cooperative editing. Although widely used for a long time, its underlying basis have not been systematically investigated. In this paper, we first conduct a systematic study of its foundations, which helps clarify its capability boundaries, practical considerations, and potential failure modes. Then, we propose a simple and elegant alternative that replaces backward spreading with forward-propagation. Instead of optimizing the target at the last editing layer, we optimize the anchor point at the first editing layer, and then propagate it forward to obtain accurate and mutually compatible target hidden-states for all subsequent editing layers. This approach achieves the same computational complexity as existing methods while producing more accurate layer-wise targets. Our method is simple, without interfering with either the computation of the initial target hidden state or any other components of the subsequent editing pipeline, and thus constituting a benefit for a wide range of LLM parameter editing methods.

## 1. Introduction

Large language models (LLMs) are known to store a substantial amount of factual and associative knowledge in their parameters. Model editing aims to modify such knowledge

---

*Equal contribution    [1]National University of Singapore [2]Laboratory for Artificial Intelligence and New Forms of Education, National Engineering Research Center for Educational Big Data, Faculty of Artificial Intelligence in Education, Central China Normal University [3]University of Oxford. Correspondence to: Zhiying Deng <zhiyingdzy@ccnu.edu.cn>.

*Proceedings of the 43rd International Conference on Machine Learning*, Seoul, South Korea. PMLR 306, 2026. Copyright 2026 by the author(s).

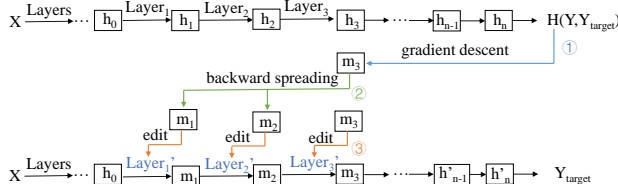

*Figure 1.* (a) A toy example of the backward spreading of ideal hidden states (i.e., $m$) in model editing. ① Getting the target hidden state of the final decisive layer by taking it as optimizable parameters and minimizing cross-entropy. ② Getting all the target hidden states of the decisive layers with backward spreading. ③ Using the target hidden states to guide the parameter editing.

post hoc. Recent work shows that editing a small subset of parameters (often within a narrow range of MLP layers) can effectively alter specific model behaviors while preserving other knowledge (Meng et al., 2022; 2023), which is called locate-then-edit model editing (referred as LTE). Owing to its favorable computational efficiency and effectiveness, LTE has emerged as one of the mainstream paradigms for LLM parameter editing.

The typical paradigm of LTE operates as follows (Meng et al., 2023). First, it precisely locates a decisive token in the input text and identifies the decisive layers in the model (usually the MLP components within several Transformer blocks). Then, if the hidden state of the decisive token at the final decisive layer can be steered toward a target hidden state (e.g, $m_3$ in Figure 1), the model will ultimately output the desired target knowledge. However, modifying a single layer is often insufficient to realize the desired steering (as the optimization objective is regularized by other regularizers). As a result, practical methods typically spread the update across multiple consecutive layers to better approach the target hidden state.

A central component of LTE is the formulation of the ideal hidden states for the decisive layers. Existing multi-layer editing strategies follow a common paradigm (Meng et al., 2023; Fang et al., 2025). They first compute the target hidden state at the last edited layer using gradient descent, and then spread it backward across earlier layers via linear interpolation, as shown with a toy example in Figure 1 (see details in Appendix A.2 and §3). In backward spreading,

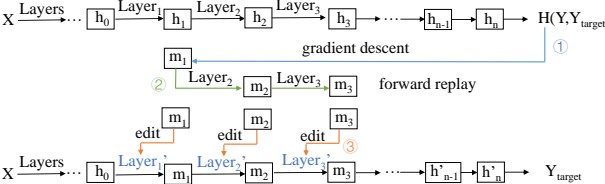

*Figure 2.* (a) A toy example of our method: forward replay of all hidden states (i.e., $m$) in model editing. ① Finding $m_1$ with gradient descent. ② Picking up the hidden states stored in the back-propagation path with forward-propagation replay. ③ Editing the model with the target hidden states.

a single residual direction (e.g., $m_3 - h_3$ in Figure 1) optimized at the final layer is broadcast to earlier layers without accounting for the layer-dependent forward dynamics. This design implicitly assumes that the residual direction computed at the final layer is consistent with the optimal update directions of all preceding layers. However, this assumption is clearly violated in general. As a consequence, the target hidden states assigned to earlier layers through backward spreading are inherently imprecise. Li et al. (2025a) propose independently performing gradient descent to solve for the ideal hidden states at multiple layers. They also acknowledge that this strategy incurs a substantial increase in computational cost. As a compromise, they reduce the number of edited layers, limiting edits to the last decisive layer and the first decisive layer (where backward spreading errors are expected to be the largest due to the longest distance to the final decisive layer). Nevertheless, this compromise introduces several notable limitations. First, even when targets are independently optimized for only two layers, the computational cost for them is doubled. Second, the target resides in a high-dimensional space and there are various permutations of it to get the same final output (Entezari et al., 2022) (more intuitively, the solution for the target hidden state is not unique, and many different input vectors can achieve the same cross-entropy loss); as a result, independently optimized targets across layers are not guaranteed to be mutually compatible. Third, restricting the number of editable layers inevitably lead to a waste of editing capacity.

Despite its widespread adoption, the basis of backward spreading has not been explicitly examined. In this work, we first conduct a systematic analysis of this issue, explaining why such methods can work in practice despite their theoretical limitations, while also clarifying their failure modes and practical considerations.

Then, to overcome this limitation, we propose a simple solution based on forward-propagation. Instead of optimizing the target hidden state at the last decisive layer, we optimize the target at the first decisive layer (Figure 2). Although only the first layer is explicitly optimized, the entire gradient backpropagation path from the final output to the first layer is implicitly incorporated into the optimization process.

Consequently, the resulting anchor point is intrinsically consistent with the downstream transformation dynamics of the model. The hidden states for all subsequent layers can therefore be easily recovered via standard forward propagation replay (see §5 and Appendix A.3), ensuring not only computational efficiency but also cross-layer compatibility.

Our method is simple and constitutes an effective improvement. It modifies only the propagation mechanism of anchor points, without interfering with either the computation of the initial target hidden state or any other components of the subsequent editing pipeline. As a result, it can hopefully be used by the vast majority of existing LTE methods.

In summary, the contributions include: (1) We systematically analyse the foundations of the current intuition-driven target construction method, formally elucidating the reasons for its effectiveness, while also clarifying its practical considerations and failure modes. (2) We propose a simple alternative based on forward propagation, which obtains mutually compatible targets across multiple layers without introducing additional computational cost. (3) Experiments[1] demonstrate that our method not only outperforms baseline approaches, but also consistently improves several recent representative methods, highlighting its scalability.

## 2. Related Works

### 2.1. LLM parameter editing

**Locate-then-edit paradigm.** Recent work has increasingly focused on precise parameter editing, typically following a *locate-then-edit* (LTE) paradigm: identifying decisive tokens and layers and directly modifying the corresponding hidden states or parameters to substitute target knowledge. Early methods such as ROME and MEMIT (Meng et al., 2022; 2023) introduced causal tracing to locate effective edit positions. Subsequent approaches, including RECT (Gu et al., 2024), EMMET (Gupta et al., 2024), PMET (Li et al., 2024), PRUNE (Ma et al., 2025), AdaEdit (Li & Chu, 2025), AlphaEdit (Fang et al., 2025), LTI (Zhang et al., 2025a),AnyEdit (Jiang et al., 2025a), and so on (Jiang et al., 2025b; Zhang et al., 2025b; Wang et al., 2025), extend this framework with improved optimization or regularization strategies. Owing to their parameter efficiency and strong empirical performance, these methods have become one of the mainstreams for knowledge editing, with recent extensions to more complex tasks such as multi-hop reasoning (Dong et al., 2025; Zhang et al., 2025a).

**Multi-layer targets spreading**. The prevailing practice in LTE methods is to first compute a target hidden state corresponding to the new knowledge at the final decisive layer. This target then serves as an anchor point, from which the

---

[1]Code: https://github.com/jugechengzi/FE

discrepancy between the current hidden state and the target hidden state is propagated backward to earlier decisive layers, thereby constructing approximate targets for those layers. Finally, multi-layer steering is applied to collectively drive the model toward the desired target knowledge (Meng et al., 2023). A recent work (Li et al., 2025a) observes that backward spreading performs suboptimally and proposes to independently compute targets for the first and the final decisive layers, editing only these two layers. However, as discussed in §1, it does not address the fundamental issue and is subject to several inherent limitations. Our paper delves more deeply into the foundations of backward spreading, elucidating the reasons for its effectiveness beyond vague intuitions, while also clarifying its practical considerations and failure modes.

### 2.2. Other methods for knowledge editing

**RAG-style inference-time knowledge editing**. A line of work draws inspiration from Retrieval-Augmented Generation (RAG), where an external knowledge base is maintained and queried during generation to provide updated information (Hartvigsen et al., 2023; Zheng et al., 2023; Zhang et al., 2024; Jiang et al., 2024; Youssef et al., 2025b). While effective in some settings, these methods inherit the limitations of RAG, including reliance on retrieval quality, context-length constraints, and increased system latency and complexity. As they do not modify model parameters, we consider them RAG rather than genuine parameter editing methods, and thus are out of the scope of this paper.

**Augmenting models with extra memory or hypernetworks.** Another stream augments LLMs with auxiliary memory modules or hypernetworks to store new knowledge. Representative methods include MEND (Mitchell et al., 2022a), KE (De Cao et al., 2021), and RLEdit (Li et al., 2025b), which often employ low-rank updates for efficiency. However, hypernetwork-based approaches typically generalize poorly, requiring retraining or finetuning for each new fact. Related methods that store knowledge in auxiliary parameters or networks, such as T-Patcher (Huang et al., 2023), SERAC (Mitchell et al., 2022b), Grace (Hartvigsen et al., 2023), WISE (Wang et al., 2024a), and KDE (Xu et al., 2025), suffer from scalability issues, as knowledge accumulates without removing outdated information, and generally are more sensitive to hyperparameters than LTE.

**Evaluation and and other aspects**. Currently, most research on evaluation and benchmarking primarily focuses on the target knowledge aspect. UniEdit (Chen et al., 2025) proposes a unified, open-domain benchmark that standardizes the evaluation of knowledge editing across diverse domains. WILD (Yang et al., 2025) argues that the editing efficacy should be evaluated with LLM as a judge. ThinkEval (Baser et al., 2025) introduces a graph-based evaluation

framework. (Youssef et al., 2025a) warn that model editing may introduce safety risks. (Youssef et al., 2025c) propose a method for detecting whether a given fact has been edited.

## 3. Preliminaries of LLM Parameter Editing

**Notation**. Let $f_\theta$ denote a pre-trained LLM parameterized by $\theta$. After editing, the updated model is denoted as $f_{\theta^*}$. We define the target knowledge set as

$$S^* = \{(q_i^*, y_i^*)\}_{i=1}^n, \tag{1}$$

where $q_i^*$ is an input query that triggers a specific piece of factual knowledge (e.g., *"The president of the US is"*), $y_i^*$ is the desired output after editing (e.g., *"Trump"*), and $n$ denotes the number of knowledge items to be updated.

To encourage the preservation of other knowledge, a representative set

$$S = \{(q_j, y_j)\}_{j=n+1}^{n+u} \tag{2}$$

is typically sampled from a background corpus such as Wikipedia. Notably, $y_j$ are obtained with $y_j = f_\theta(q_j)$, rather than really sampled from Wikipedia.

**General editing objective**. The overall goal of knowledge editing is to update specific factual associations while minimally affecting the model's behavior on other inputs. This objective can be formulated as:

$$\theta^* = \arg\min_{\hat{\theta}} \Big( \sum_{i=1}^n \mathcal{L}_1(f_{\hat{\theta}}(q_i^*), y_i^*)$$
$$+ \lambda \sum_{j=n+1}^{n+u} \mathcal{L}_2(f_{\hat{\theta}}(q_j), y_j) \Big), \tag{3}$$

where $\mathcal{L}_1$ and $\mathcal{L}_2$ denote loss functions for enforcing the target edit and preserving other knowledge, respectively, and $\lambda$ controls the trade-off between them.

**Locate-then-edit paradigm**. To do editing more efficiently, prior work proposes the *locate-then-edit* (LTE) paradigm (Meng et al., 2022; 2023) that gives a succinct solution for it. LTE first identifies where a piece of factual knowledge is stored in the model and then apply targeted parameter updates. A widely adopted approach is *causal tracing* (Meng et al., 2022), which identifies:

- **Decisive token**: the token whose hidden representation most strongly determines the factual output, often corresponding to the final token of the subject entity in the input.

- **Decisive layers**: the model layer at which intervening on the hidden state of the decisive token most effectively steers the model toward the desired edit target, often corresponding to several MLP layers.

By intervening only on the hidden state of the decisive token at the decisive layer, these methods can selectively overwrite specific factual associations with high efficiency and precision. Due to its simplicity and strong empirical performance, this paradigm has become one of the mainstreams in the knowledge editing literature.

Despite differences in implementation, many locate-then-edit methods can be expressed as variants of the following optimization problem (Meng et al., 2023):

$$\Delta_W^* = \arg\min_{\Delta_W}\left(\sum_{i=1}^n \|(W + \Delta_W)k_i - m_i\|^2 + \lambda \sum_{j=n+1}^{n+u} \|\Delta_W k_j\|^2\right), \tag{4}$$
$$W^* = W + \Delta_W^*,$$

where $W$ denotes a selected subset of model parameters to be edited, $k_i$ is the hidden representation of the decisive token at the layer immediately preceding $W$, and $m_i$ is an idealized hidden representation aligned with the desired edit target. The second regularization term is designed to enforce orthogonality between $\Delta_W$ and other non-target token embeddings $k_j$. The target $m_i$ is obtained by treating it as an optimizable parameter and applying gradient descent to minimize the cross-entropy between the current model output and the target output (see Appendix A.1 for details).

Eq. 4 admits a closed-form solution:

$$\Delta_W^* = (M_I - WK_I)K_I^\top(K_I K_I^\top + \lambda K_J K_J^\top)^{-1}, \tag{5}$$

where $M_I, K_I, K_J$ represent the concatenations of all $m_i$, $k_i$, and $k_j$, respectively. In practice, we first take $m_i$ as optimizable parameters and use gradient descent to get it. Then, we apply $M_I$ to Eq 5 to get $\Delta_W^*$.

**Steering the hidden state with multi-layer cooperation**. It is difficult to directly steer the model output to the desired target hidden state $m$ through a parameter edit at a single layer (as it is regularized by preserving non-target knowledge), and thus model editing is typically implemented by sequentially modifying multiple layers. For multi-layer editing, a straightforward approach is to separately compute a layer-specific $m$ for each layer and sequentially apply Eq. 5. However, this strategy leads to a significant increase in the computational cost of solving for $m$. As a result, the prevailing practice is to compute $m$ only at the last layer to be edited, and then spreading the discrepancy

$$r = m - h, \quad s.t., \ h = Wk. \tag{6}$$

across multiple layers via interpolation to get the target hidden states of other layers (e.g., in the example of Figure 1 we may have $m_1 = h_1 + \frac{m_3 - h_3}{3}$) (Meng et al., 2023), as shown with a detailed example in Appendix A.2.

## 4. How does spreading work

Although backward spreading is widely used in practice, its theoretical grounds and underlying basis have not been systematically investigated. In this section, we conduct a systematic study of its foundations, which helps clarify its capability boundaries, practical considerations, and potential failure modes.

### 4.1. Theoretical grounding

**Notations**. Consider that we are editing $L$ layers $W_1, W_2, ...W_L$ (note that the decisive layers are MLP layers). Let $k_l$ denote the input of the decisive token before the $l$-th layer. Let $\delta_{m_l}$ denote the change in output of layer-$l$, i.e., $\delta_{m_l} = \Delta_{W_l} k_l$. Let $\delta_{m_L}^l$ denotes the passive change at layer-$L$ induced by an intervention $\delta_{m_l}$ applied at layer-$l$ (i.e., the superscript means passive change). We consider that $\delta_{m_l}$ is a small value (i.e., first-order approximation can be applied), we have

$$\delta_{m_L}^l = J_{l \to L} \delta_{m_l}, \tag{7}$$

where $J_{l \to L}$ (a square matrix) denotes the Jacobian mapping perturbations in $W_l$ output to $W_L$ output.

The initial total residual at the final layer-$L$ is iteratively interpolated from shallow to deep layers (see Appendix A.2) to get the target hidden state for all decisive layers. Consider that we now want to edit $W_l$, and the current remaining total residual (after editing layer $W_{l-1}$) at layer-$L$ is $R_L^{l-1}$. Then, the residual assigned to layer-$l$ is $\frac{R_L^{l-1}}{L-l+1}$. That's to say, the target steering intervention at layer-$l$ is

$$\delta_{m_l} = \frac{R_L^{l-1}}{L - l + 1}. \tag{8}$$

Ideally, we want $\delta_{m_L}^l$ to be in the same direction of current remaining residual $R_L^{l-1}$, that is to say,

$$\delta_{m_L}^l = \alpha R_L^{l-1} = \beta \frac{R_L^{l-1}}{L - l + 1} = \beta \delta_{m_l} \tag{9}$$

To make the backward residual spreading work, ideally, we need to have

$$\forall l < L, \quad \exists 0 < \beta, \quad \delta_{m_L}^l = \beta \delta_{m_l}. \tag{10}$$

In fact, this involves a strong assumption:

**Theorem 1.** *Eq. 10 holds if and only if $\delta_{m_l}$ is an eigenvector of $J_{l \to L}$, with $\beta$ being the corresponding eigenvalue.*

The proof is trivial. Please see Appendix A.6.

**Remark**. We know that $\delta_{m_l}$ is aligned with $R_L^{l-1}$, whose direction is jointly determined by both the layers after $W_L$

*Table 1.* The cosine similarity between $\delta_{m_l}$ and $\delta_{m_L}^l$.

| Layers | Layer4 | Layer5 | Layer6 | Layer7 | Layer8 |
|--------|--------|--------|--------|--------|--------|
| Cosine | 0.34 | 0.41 | 0.54 | 0.72 | 1.00 |

(used to obtain $m$ in Eq. 6) and the layers before it (used to obtain $k$ in Eq. 6). In contrast, $J_{l \to L}$ is determined solely by the layers between $l$ and $L$. Since these two quantities are formed through very different mechanisms, there is no inherent reason to expect such a special relationship between them. Therefore, requiring $\delta_{m_l}$ to be an eigenvector of $J_{l \to L}$ constitutes a very strong condition.

However, in practice, we find that the backward residual spreading manner still works to some extent. So, what is the reason? In practice, although $\delta_{m_L}^l$ may not be in the same direction with $\delta_{m_l}$, it may nevertheless move toward it along a indirect trajectory. In such cases, it suffices that

$$\cos(\delta_{m_L}^l, \delta_{m_l}) > 0,$$

which is based on a much milder condition:

**Theorem 2.** *If the symmetric part of the Jacobian mapping perturbations $J_{l \to L}$ is positive definite, then the induced perturbation at layer $L$ resulting from an intervention at layer $l$ does not oppose the target residual:*

$$\frac{J_{l \to L} + J_{l \to L}^\top}{2} > 0 \implies \cos(\delta_{m_L}^l, \delta_{m_l}) > 0. \quad (11)$$

The proof is in Appendix A.7.

**Remark.** $\frac{J_{l \to L} + J_{l \to L}^\top}{2} > 0$ is not guaranteed to hold for arbitrary inputs or directions. Nevertheless, in practical settings involving modern deep neural networks trained with standard optimization and regularization techniques, this condition is observed to hold in a large fraction of cases. This is because that robust networks implicitly favor smoother and more stable input–output mappings, while strongly reversing or highly rotational Jacobian behaviors are discouraged due to stability and generalization considerations (Keskar et al., 2017; Chizat & Bach, 2018). Moreover, regularization techniques such as weight decay and normalization layers further promote locally well-conditioned Jacobians (Ross & Doshi-Velez, 2018). However, it is worth noting that increasing the depth between two layers generally introduces stronger rotational effects in the corresponding Jacobian, which in turn reduces the likelihood that this condition holds.

**Empirical verification**. We subsequently conduct real-world experiments to empirically validate the above analysis and discussion. We adopt the standard MEMIT (Meng et al., 2023) editing method and perform experiments on the Llama3-8B-Instruct model (the decisive layers are $[4, 5, 6, 7, 8]$). Edits are performed on 2,000 samples from the CounterFact dataset (Meng et al., 2023). For each decisive layer $l$, we examine the cosine similarity between the

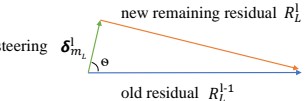

*Figure 3.* The steering step should be small if $\theta$ is large.

passive shift induced at the final layer $L$ (i.e., $\delta_{m_L}^l$) after applying steering at layer $l$ (i.e., $\delta_{m_l}$).

The results are in Table 1. We observe that the Jacobian $J_{l \to L}$ progressively induces larger angular deviations for layers that are farther away from layer $L$. This indicates that, under a backward spreading scheme, incorporating additional earlier layers leads to increasingly divergent effects. Such non–closed-form convergence behavior is unfavorable for scalability (e.g., it suggests a low upper bound on the number of layers that can be reliably edited using this approach).

**Be careful about the step size**. A natural follow-up question is whether backward spreading can be safely applied even if the condition $\cos(\delta_{m_L}^l, R_L^{l-1}) > 0$ is satisfied, or whether additional considerations are still required.

We note that, when steering under indirectly assigned targets obtained via backward spreading, each update step should be taken cautiously and kept sufficiently small because we don't know where the real target is located. As shown in Figure 3, once $\|\delta_{m_L}^l\|$ is too large, the steering update can even move we farther away from the target rather than closer to it. But if it is too small, the gain will also be small. As a result, we must carefully control $\delta_{m_l}$ to make $\|\delta_{m_L}^l\|$ small enough (but note that $\|\delta_{m_L}^l\|$ is a passive variable and is not directly controllable under the backward spreading manner).

**Empirical verification**. The prevailing practice performs editing sequentially from shallow to deep layers, constructing the steering vector at each layer by dividing the remaining residual at the final layer by the number of layers yet to be edited. To assess the contribution of each layer, we measure the ratio between the norm of the residual remaining after editing a given layer and the norm of the original total residual (the residual is taken from the final decisive layer). In addition, we compare this strategy with an alternative in which the steering vector is constructed directly from the remaining residual without normalization by the number of remaining editable layers. We also add a baseline OneLayer (only the last decisive layer is edited) for comparison. The model and data are the same as those used in Table 1. Please see Appendix A.5 for more details.

The results are in Table 2. First, we examine the results of OneLayer and observe that it is indeed difficult to directly steer the model to the target through single-layer editing, indicating that multi-layer cooperation is necessary. We then examine the "Dividing" setting, in which the remaining total residual is normalized by the number of layers yet to be

*Table 2.* The ratio of remaining final residual after editing each decisive layer. The model is Llama3-8B-Instruct.

| Layers | Layer4 | Layer5 | Layer6 | Layer7 | Layer8 |
|---|---|---|---|---|---|
| Dividing | 0.96 | 0.91 | 0.83 | 0.69 | 0.36 |
| No-dividing | 1.19 | 0.90 | 0.74 | 0.56 | 0.23 |
| OneLayer | - | - | - | - | 0.55 |

edited. Although this normalization is intended to enforce equal contributions from each layer, the severe directional distortion causes the contributions of the earlier layers to be negligible in practice (particularly for the first two layers). This observation is consistent with the findings reported by Li et al. (2025a). We then consider the "No-dividing" setting, where no such normalization is applied. While this variant achieves a lower final residual at Layer8, the steering at Layer4 paradoxically moves the representation farther away from the target. This behavior empirically confirms the difficulty of controlling the update step size.

More critically, since the residual is propagated backward from Layer8, the results reveal that the effectiveness of adding earlier editable layers progressively degrades. In other words, increasing the number of editable layers does not promise the convergence of the remaining residual. This non-convergent behavior severely limits the scalability of backward spreading–based LLM editing.

## 5. Our method

From the preceding analysis, we learn that backward residual spreading fails to provide accurate edit targets for layers preceding the final decisive layer. Consequently, we need a principled alternative for constructing layer-wise edit targets. A natural intuition is that we can compute the target $m$ of Eq. 4 at each layer separately. However, this will significantly increase computational cost, rendering it impractical for modern LLMs. Moreover, since $m$ is a high-dimensional vector, solving for it via gradient-based optimization does not yield a unique solution. As a result, independently computed targets across multiple layers may be mismatched and lack cross-layer compatibility. In this section, we introduce a more efficient formulation that reshapes multi-layer parameter editing by replacing backward residual spreading with forward-propagation. Our method achieves accurate layer-wise edit targets with the same computational complexity as existing approaches, while respecting the true forward dynamics of the network.

Our method is illustrated in Figure 2. First, following standard MEMIT (Meng et al., 2023), we perform a single gradient-descent optimization. The key difference lies in the choice of the optimizable variable: whereas MEMIT treats the hidden state at the final decisive layer as the optimizable parameter, we instead optimize the hidden state at the first decisive layer. After obtaining the target hidden state at the first decisive layer, we use it as input and recover the target hidden states at the corresponding positions of all subsequent layers via standard forward propagation (see Appendix A.3 for details and Appendix A.4 for the theoretical explanation). With the exception of the second stage, all other components remain identical to the standard LTE method MEMIT. The motivation is clear and straightforward: when solving for the ideal target hidden state $m_1$ via backpropagation, the ideal states of the subsequent layers that are consistent with $m_1$ are already implicitly encoded along the backward computation path. By simply recovering these states through standard forward propagation, we can obtain accurate and mutually compatible hidden states for the remaining layers.

Importantly, this procedure does not interfere with any other stage of the editing pipeline; it only provides more accurate editing targets at each layer. As a result, a wide range of existing LTE methods can benefit from this approach with minimal modification.

## 6. Experiments

### 6.1. Setup

**Baselines**. We call our method FE (**f**orward propagation **e**dit), which is built upon the standard MEMIT (Meng et al., 2023) framework by replacing backward spreading with forward propagation. We first construct several baselines in a controlled manner to closely examine the effectiveness of our method. (1) Standard MEMIT (Meng et al., 2023). This baseline follows the original MEMIT procedure, using backward spreading to construct targets for earlier decisive layers from the target computed at the final decisive layer. (2) OneLayer. Only the final decisive layer is edited. (3) BLUE (Li et al., 2025a). Gradient-based optimization is independently performed twice to obtain targets for the first and the final decisive layers, and only these two layers are edited. Note that BLUE requires independently computing targets for two layers, thereby doubling the computational cost. Since our method modifies only the target propagation mechanism without interfering with other components of the editing pipeline, it is expected to benefit a wide range of editing methods. Accordingly, we further extend our approach to several recently proposed methods to evaluate its scalability and general applicability. These methods include RECT (Gu et al., 2024), Alphaedit (Fang et al., 2025) and PRUNE (Ma et al., 2025). We also include two methods that do not belong to the LTE paradigm: WISE (Wang et al., 2024a) and RLEdit (Li et al., 2025b). WISE introduces additional memory modules that are trained to store and retrieve new knowledge, whereas RLEdit updates knowledge by training a hypernetwork to generate parameter modifications.

**Models, datasets, and metrics**. Following the prevailing experimental settings in the model editing literature, we

*Table 3.* Effectiveness of our FE. The model is Llama3-8b-instruct.

| | The MCF dataset | | | | | | | |
| Metrics | Efficacy | | Generalization | | Specificity | | | |
| Methods | Success ↑ | Accuracy ↑ | Success ↑ | Accuracy ↑ | $D_{KL}$ ↓ | Top-1 ↑ | Top-5 ↑ | Top-10 ↑ |
|---|---|---|---|---|---|---|---|---|
| OneLayer | 76.2 | 59.7 | 48.3 | 21.0 | 0.28 | 80.5 | 77.4 | 76.5 |
| MEMIT | 97.4 | 94.3 | 78.7 | 48.0 | 0.41 | 77.5 | 75.2 | 74.4 |
| BLUE | 98.4 | 96.9 | 90.0 | 60.1 | 0.35 | 81.0 | 77.8 | 77.2 |
| WISE | 18.0 | 2.3 | 11.8 | 0.8 | **0.09** | **97.7** | **97.5** | **97.4** |
| RLEdit | 57.8 | 23.1 | 51.5 | 17.8 | 5.62 | 25.7 | 8.6 | 9.8 |
| PRUNE | 97.4 | 94.4 | 78.7 | 47.9 | 0.43 | 76.8 | 74.8 | 73.8 |
| RECT | 96.9 | 93.4 | 77.7 | 46.7 | 0.42 | 76.9 | 75.1 | 74.1 |
| Alphaedit | 95.2 | 89.0 | 76.3 | 44.4 | 0.63 | 71.2 | 68.9 | 67.9 |
| MEMIT+FE | **99.9** | **99.8** | **93.0** | **61.0** | 0.34 | 82.7 | 78.8 | 77.8 |

| | The ZsRE dataset | | | | | | | |
| Metrics | Efficacy | | Generalization | | Specificity | | | |
| Methods | Success ↑ | Accuracy ↑ | Success ↑ | Accuracy ↑ | $D_{KL}$ ↓ | Top-1 ↑ | Top-5 ↑ | Top-10 ↑ |
|---|---|---|---|---|---|---|---|---|
| OneLayer | 83.9 | 67.2 | 79.2 | 59.0 | 0.71 | 43.3 | 65.1 | 69.5 |
| MEMIT | 92.6 | 82.0 | 87.6 | 72.6 | 0.60 | 45.3 | 67.0 | 71.9 |
| BLUE | 94.6 | 85.6 | 90.9 | 78.2 | 0.18 | 66.7 | 81.3 | 83.7 |
| WISE | 40.6 | 2.0 | 39.3 | 1.6 | 1.75 | 53.1 | 57.8 | 57.5 |
| RLEdit | 71.0 | 47.6 | 69.2 | 44.0 | 3.51 | 1.7 | 8.5 | 15.1 |
| PRUNE | 92.6 | 81.9 | 87.7 | 72.7 | 0.59 | 46.2 | 66.9 | 71.7 |
| RECT | 92.1 | 80.9 | 87.1 | 71.8 | 0.58 | 46.6 | 67.2 | 71.8 |
| Alphaedit | 90.8 | 77.8 | 85.8 | 69.4 | 0.92 | 32.6 | 56.9 | 63.8 |
| MEMIT+FE | **97.6** | **93.0** | **93.6** | **84.3** | **0.09** | **75.6** | **86.9** | **88.2** |

evaluate our method on three LLMs: LLaMA3-8B-Instruct, GPT-J, and GPT-2 XL. Among them, LLaMA3-8B-Instruct more closely reflects the characteristics of contemporary mainstream LLMs, and is therefore used as our primary evaluation model. Experiments on GPT-J and GPT-2 XL are reported in Appendix A.8 (the findings are generally consistent with those observed on LLaMA3-8B-Instruct). We follow the recent work Alphaedit in using two benchmark datasets: Multi-CounterFact (MCF) (Meng et al., 2022) and ZsRE (Levy et al., 2017), editing 2,000 knowledge items for each dataset. Consistent with standard MEMIT practice, we perform massive editing, in which all knowledge items are edited jointly in a single batch. All hyperparameters, including the selection of decisive layers, are adopted from EasyEdit (Wang et al., 2024b), a well-known public repository widely used in model editing community. Following common practice, we report metrics along three dimensions: efficacy, generalization, and specificity. These metrics jointly provide a comprehensive evaluation of editing quality. Efficacy assesses whether the intended knowledge has been correctly injected, while generalization evaluates the robustness of the edit under natural linguistic variations, reflecting whether the model has internalized the knowledge rather than memorized a specific query. Specificity measures the extent to which the edit preserves unrelated model behavior, which is critical for practical deployment. Together, these metrics capture the essential trade-offs in

knowledge editing between correctness, robustness, and minimal unintended side effects. Both efficacy and generalization are assessed using two metrics: **Success** (Meng et al., 2023) and **Accuracy** (Yang et al., 2025). *Success* follows the traditional teacher-forcing–based evaluation widely used in the literature, whereas *Accuracy* measures exact token match under free-form generation. For specificity, prior work by Liu et al. (2026) has shown that conventional ground-truth–based specificity metrics fail to accurately capture behavioral changes induced by editing. Accordingly, we follow Liu et al. (2026) and adopt more accurate ground-truth–free behavior deviation metrics. Specifically, $D_{KL}$ measures the KL-divergence between output representations before and after editing, while Top-$k$ quantifies the overlap ratio of the top-$k$ supported outputs (Liu et al., 2026).

### 6.2. Results

**Comparison with baseline methods.** The results of the controlled baselines are reported in Table 3. Compared to the standard MEMIT, OneLayer performs poorly, indicating that directly steering the model to the target by editing a single layer is challenging. When comparing BLUE with OneLayer, although the improvement in efficacy is intuitive, an interesting observation is that BLUE sometimes achieves better specificity despite editing more layers. This phenomenon may be attributed to the fact that concentrating the

*Table 4.* Scalability of our FE. The model is Llama3-8b-instruct.

| | The MCF dataset | | | | | | | |
|---|---|---|---|---|---|---|---|---|
| Metrics
Methods | Efficacy | | Generalization | | Specificity | | | |
| | Success ↑ | Accuracy ↑ | Success ↑ | Accuracy ↑ | $D_{KL}$ ↓ | Top-1 ↑ | Top-5 ↑ | Top-10 ↑ |
| PRUNE | 97.4 | 94.4 | 78.7 | 47.9 | 0.43 | 76.8 | 74.8 | 73.8 |
| PRUNE+FE | 99.9 | 99.8 | 93.0 | 61.1 | 0.34 | 82.7 | 78.8 | 77.8 |
| RECT | 96.9 | 93.4 | 77.7 | 46.7 | 0.42 | 76.9 | 75.1 | 74.1 |
| RECT+FE | 99.9 | 98.0 | 92.6 | 60.4 | 0.34 | 83.0 | 78.9 | 77.9 |
| Alphaedit | 95.2 | 89.0 | 76.3 | 44.4 | 0.63 | 71.2 | 68.9 | 67.9 |
| Alphaedit+FE | 99.8 | 99.2 | 92.2 | 59.1 | 0.47 | 77.1 | 72.7 | 71.4 |

| | The ZsRE dataset | | | | | | | |
|---|---|---|---|---|---|---|---|---|
| Metrics
Methods | Efficacy | | Generalization | | Specificity | | | |
| | Success ↑ | Accuracy ↑ | Success ↑ | Accuracy ↑ | $D_{KL}$ ↓ | Top-1 ↑ | Top-5 ↑ | Top-10 ↑ |
| PRUNE | 92.6 | 81.9 | 87.7 | 72.7 | 0.59 | 46.2 | 66.9 | 71.7 |
| PRUNE+FE | 97.6 | 93.0 | 93.6 | 84.4 | 0.09 | 74.3 | 86.7 | 88.2 |
| RECT | 92.1 | 80.9 | 87.1 | 71.8 | 0.58 | 46.6 | 67.2 | 71.8 |
| RECT+FE | 97.6 | 93.1 | 93.6 | 84.4 | 0.09 | 74.1 | 86.8 | 88.0 |
| Alphaedit | 90.8 | 77.8 | 85.8 | 69.4 | 0.92 | 32.6 | 56.9 | 63.8 |
| Alphaedit+FE | 96.6 | 90.5 | 92.2 | 81.8 | 0.19 | 69.3 | 82.7 | 84.4 |

*Table 5.* The ratio of remaining final residual after editing each decisive layer. The model is Llama3-8B-Instruct.

| Layers | Layer4 | Layer5 | Layer6 | Layer7 | Layer8 |
|---|---|---|---|---|---|
| FE | 0.37 | 0.15 | 0. 08 | 0.05 | 0.03 |

primary editing effort on shallower layers can be beneficial for specificity. Finally, we observe that our method significantly outperforms the standard MEMIT baseline, as well as the improved BLUE variant and all other baselines. Notably, BLUE incurs substantially higher computational cost than other methods, whereas our FE approach has nearly the same computational cost as standard MEMIT. This is because the additional forward propagation involves only four layers, rendering the overhead effectively negligible.

**Scalability**. To demonstrate the generality and scalability of FE, we further extend our method to several representative methods. The results are in Table 4. We find that these methods also benefit to some extent from simply replacing backward spreading with our proposed forward replay.

**Convergent property**. In backward spreading, the final decisive layer is treated as the anchor point, and additional editable layers are progressively included toward earlier layers. The experiments reported in Table 2 show that the marginal gains diminish rapidly as the propagation spans farther layers, indicating that increasing the number of backward-spread layers does not guarantee convergence of the discrepancy between the target at the final decisive layer and the actual hidden state. We conduct the same set of experiments in Table 2 for our method to examine the convergence behavior of this discrepancy as the number of editing layers increases.

The results are in Table 5. Our method starts from the first decisive layer and obtains the targets for subsequent layers through forward propagation. As shown in the results, the discrepancy decreases approximately exponentially as more subsequent layers are included (roughly halving at each layer), demonstrating good convergence behavior. Consequently, the final error is substantially lower than that of the two backward-spreading–based methods reported in Table 2, with the discrepancy at Layer8 being less than one-tenth of theirs. Moreover, this favorable convergence property enables flexible adjustment of the number of edited layers to meet different error tolerance requirements. In addition, as discussed earlier in the comparison between BLUE and OneLayer, concentrating the primary editing effort on shallower layers may better preserve specificity. This constitutes another advantage of our approach.

## 7. Conclusion and Limitations

This work reshapes the acquisition of multi-layer targets from the traditional intuition-based backward-spreading approach into a more principled forward-propagation–based replay scheme. This reformulation yields substantially more accurate multi-layer targets while incurring virtually no additional computational overhead. One remaining limitation is that the primary editing effort is concentrated in earlier layers. Designing more effective coordination strategies across layers, such that editing demands can be distributed in a principled manner, remains an open problem for future work. Nevertheless, existing methods existing methods likewise fail to achieve effective layer-wise allocation. Compared to current strategies, this study already represents a considerable step forward: although it does not yet realize

perfectly coordinated layer-wise editing, it provides each layer with a more accurate target and, importantly, exhibits error convergence as the number of edited layers increases. Empirically, it also delivers substantial improvements across efficacy, generalization, and specificity.

## Impact Statement

This paper presents work whose goal is to advance the field of Machine Learning. There are many potential societal consequences of our work, none which we feel must be specifically highlighted here.

## Acknowledgments

The research is supported by the National Research Foundation, Singapore, and the Ministry of Digital Development and Information (MDDI) under the Singapore Global AI Visiting Professorship Program (Award No. AIVP-2024-002). It is also supported by the National Natural Science Foundation of China (62507018).

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

# A. More Details

## A.1. How to get the target hidden state $m$

In Eq. 4, a critical step is to obtain the target hidden state $m$. The prevailing approach treats $m$ as an optimizable variable and solves for it by minimizing the cross-entropy loss between the model's current output and the target answer via gradient descent.

Concretely, suppose we are given a query token sequence

$$Q = [q_1, q_2, ..., q_d, ..., q_n],$$ (12)

where $q_d$ denotes the decisive token, and a target answer sequence

$$Y = [y_1, y_2, y_3]$$ (13)

Let the final decisive layer be layer-$l$. After passing the concatenated sequence $[Q, Y]$ through the first $l$ layers of the LLM, we obtain a sequence of hidden states $h(Q, Y) = [h_{q_1}, ..., h_{q_d}, ...h_{q_n}, ...h_{y_3}]$. We denote the remaining layers of the model by a function $\hat{Y} = f(h(Q, Y))$. The cross-entropy loss, computed only over the target answer tokens, is denoted by $H_c(Y, \hat{Y})$. By treating the hidden representation of the decisive token $h_{q_d}$ as the optimizable parameter and minimizing $H_c$ via gradient descent, we obtain the optimal hidden state for $q_d$, which is taken as the target $m$.

Prior literature has shown that if the MLP parameters can be modified such that the hidden state $h_{q_d}$ is replaced with $m$, the model will ultimately generate the target answer (Meng et al., 2022; 2023). Note that the second regularization term in Eq.4 is designed to enforce orthogonality between $\Delta_W$ and other non-target token embeddings (i.e., $\Delta_W$ only changes $h_{q_d}$).

## A.2. An example about backward spreading

**Notations**. To avoid excessive subscripts and superscripts, we use subscripts to denote layer indices and superscripts to indicate the values of variables after the $i$-th layer has been edited. The symbols $k, h, m$ all denote hidden states. Specifically, we use $k$ to represent the input hidden state to an MLP, $h$ to denote the output hidden state after the MLP transformation, and $m$ to represent the target hidden state.

We consider there are five decisive layers, denoted by $[W_1, W_2, W_3, W_4, W_5]$. Note that the decisive layers are MLPs. Let $m_l$ denote the editing target computed at the output position of the $l$-th layer. And let $k_l$ denote the input of the decisive token before the $l$-th layer.

We denote $h_i = W_i k_i$, where $W_i k_i$ is the $i$-th layer's output before editing. Ideally, according to Eq. 4, the editing target at each layer can be viewed as a residual $m_i - h_i$. Then, we can rewrite the first term of Eq. 4 as:

$$\Delta_l^* = \arg\min_{\Delta_l} \|\Delta_l k_l - (m_l - h_l)\|^2$$ (14)

In practice, editing is performed sequentially from shallow to deep layers. To avoid the computational overhead of solving for a separate $m_l$ at each layer, the total residual is computed only at the final layer and then distributed across the preceding layers to be jointly resolved. Specifically, we first compute $m_5$, from which the initial total residual is given by $m_5 - W_5 k_5$.

When editing $W_1$, the residual is uniformly allocated as $\frac{(m_5 - h_5)}{5}$, which means the target hidden state is

$$m_1 = h_1 + \frac{(m_5 - h_5)}{5}.$$ (15)

And $W_1$ is edited with

$$\Delta_1^* = \arg\min_{\Delta_1} \|(W_1 + \Delta_1)k_1 - m_1\|_2,$$ (16)

It yields the updated parameter for the 1-th layer:

$$W_1^1 = W_1 + \Delta_1^*,$$ (17)

where the superscript means the result after editing the 1-st layer. After editing the first layer, $k_l$ in the following layers will also change to $k_l^1$ accordingly. Also, $h_l$ becomes $h_l^1$

Suppose the $h_5$ at the final layer becomes $h_5^1$. The remaining total residual is then $m_5 - h_5^1$, which must be jointly resolved by the **remaining four** layers yet to be edited. According, the target hidden state of $W_2$ is given by

$$m_2 = h_2^1 + \frac{(m_5 - h_5^1)}{4}. \tag{18}$$

And $W_2$ is updated with

$$\Delta_2^* = \arg\min_{\Delta_2} \|(W_2 + \Delta_2)k_2^1 - m_2\|^2, \tag{19}$$

The same procedure is applied iteratively to the remaining layers.

We call the direction of the residual (e.g., $m_2 - h_2^1 = \frac{m_5 - h_5^1}{4}$) as the steering direction at each layer.

Although this strategy is widely adopted, there has been little investigation into why it works. In fact, the inherent nonlinearity of neural networks may cause the ideal steering directions at different layers to differ substantially, making such an interpolative backward spreading far from a natural or theoretically grounded choice. Nevertheless, empirical results suggest that this method works to a certain extent in practice. Understanding why it works at all is therefore essential for clarifying its potential failure modes.

### A.3. The details about how to implement our method

Most parts of our method are the same as those in Appendix A.1.

Concretely, suppose we are given a query token sequence

$$Q = [q_1, q_2, ..., q_d, ..., q_n], \tag{20}$$

where $q_d$ denotes the decisive token, and a target answer sequence

$$Y = [y_1, y_2, y_3] \tag{21}$$

Let the **first** decisive layer be layer-$l$. After passing the concatenated sequence $[Q, Y]$ through the first $l$ layers of the LLM, we obtain a sequence of hidden states $h(Q, Y) = [h_{q_1}, ..., h_{q_d}, ...h_{q_n}, ...h_{y_3}]$. We denote the remaining layers of the model by a function $\hat{Y} = f(h(Q, Y))$. The cross-entropy loss, computed only over the target answer tokens, is denoted by $H_c(Y, \hat{Y})$. By treating the hidden representation of the decisive token $h_{q_d}$ as the optimizable parameter and minimizing $H_c$ via gradient descent, we obtain the optimal hidden state for $q_d$, which is taken as the target $m_1$.

Then we consider there are five decisive layers, denoted by $[W_1, W_2, W_3, W_4, W_5]$. Currently, we have the target $m_1$ for the first layer $W_1$. We denote the LLM submodule mapping the output of $W_1$ to the output of $W_2$ as $f_2$, and analogously define $f_3, f_4, f_5, ..., f_n$ for the subsequent layers.

We have

$$f_n \circ f_n - 1 \circ \cdots \circ f_2(h_{q_1}, ..., h_{q_d}, ...h_{q_n}, ...h_{y_3}) = \hat{Y} \tag{22}$$

By replacing $h_{q_d}$ with $m_1$, we obtain the target output $Y$ (see Appendix A.1):

$$f_n \circ f_n - 1 \circ \cdots \circ f_2(h_{q_1}, ..., m_1, ...h_{q_n}, ...h_{y_3}) = Y \tag{23}$$

Based on this formulation, we can readily recover the target hidden states of subsequent layers that are compatible with $m_1$. For example, let

$$h_2 = f_2(h_{q_1}, ..., m_1, ...h_{q_n}, ...h_{y_3}) \tag{24}$$

then the hidden state at the position corresponding to $q_d$ in $h_2$ is taken as the target $m_2$.

### A.4. Theoretical explanation of our method

The effectiveness of FE is straightforward, which is why we omitted a formal proof in the main part of this paper.

Suppose the input is $X = [x_1, x_2, ..., x_d, ..., x_n]$, where $x_d$ is the decisive token, with label Y, and let a three-layer neural network be defined as $f_3(f_2(f_1(X))) = \hat{Y}$. By gradient descent, we find an optimal $x_d = m_1$ such that $f_3(f_2(f_1(x_1, x_2, ..., m_1, ..., x_n))) = Y^*$, where $Y^*$ denotes a desirable output that yields a small cross-entropy $H_c(Y, Y^*)$.

Denote $H_1 = f_1(x_1, x_2, ..., m_1, ..., x_n) = [h_1, h_2, ..., h_d, ..., h_n]$. Then it follows immediately that $f_3(f_2(h_1, h_2, ..., h_d, ..., h_n)) = Y^*$, simply by the basic property of functions: the same input cannot produce two different outputs.

If we then take $h_d$ and define it as $m_2$, i.e., $m_2 = h_d$, it follows directly that $f_3(f_2(h_1, h_2, ..., m_2, ..., h_n)) = Y^*$. Therefore, once the edit target $m_1$ for the first layer is known, we can directly and exactly obtain the edit target $m_2 = h_d$ for the second layer through a strictly accurate forward computation. This is the theoretical basis of FE.

Since this argument is quite direct, we did not elaborate on it in detail in the main text.

### A.5. The details of the experiments in Table 2

Here are the details of the experiments in Table 2. "Dividing": the targets for each layer are got in the way of Appendix A.2. "No-Dividing": the steering vector is constructed directly from the remaining residual without normalization by the number of remaining editable layers. For example, we revise Eq. 15 to be

$$m_1 = W_1 k_1 + (m_5 - W_5 k_5). \tag{25}$$

Similarly, we revise Eq. 18 to be

$$m_2 = W_2 k_2^1 + (m_5 - W_5 k_5^1). \tag{26}$$

The values in the table mean the ratio between the norm of the residual remaining after editing a given layer and the norm of the original total residual (the residual is taken from the final decisive layer). For example, the original total residual is $m_5 - W_5 k_5$. After editing the first layer (i.e., Layer4 in Table 2), the hidden state of the final layer becomes $k_5^1$, and the remaining residual becomes $m_5 - W_5 k_5^1$. Then the value corresponding to Layer4 is $\frac{\|m_5 - W_5 k_5^1\|_2}{\|m_5 - W_5 k_5\|_2}$.

### A.6. The Proof of Theorem 1

The proof is trivial. We can get it with the definition of eigenvalue and eigenvector. We only need to rewrite Eq. 10 as

$$J_{l \to L} \delta_{m_l} = \delta_{m_L}^l = \beta \delta_{m_l}, \tag{27}$$

where the left part is from Eq. 7. Then, we can naturally get Theorem 1 with the definition of eigenvalue and eigenvector.

### A.7. The Proof of Theorem 2

We have

$$\delta_{m_L}^l = J_{l \to L} \delta_{m_l}. \tag{28}$$

Then, we want to have

$$\cos(\delta_{m_L}^l, \delta_{m_l}) > 0, \tag{29}$$

which is equal to that

$$\delta_{m_l}^\top \cdot \delta_{m_L}^l > 0, \tag{30}$$

which is equal to

$$\delta_{m_l}^\top J_{l \to L} \delta_{m_l} > 0. \tag{31}$$

We denote

$$J_{l \to L} = S + A, \quad S = \frac{J_{l \to L} + J_{l \to L}^\top}{2}, \quad A = \frac{J_{l \to L} - J_{l \to L}^\top}{2}, \tag{32}$$

where we have

$$S = S^\top, \quad A = -A^\top. \tag{33}$$

*Table 6.* The effectiveness of our FE. The model is GPT2-XL.

| | The MCF dataset | | | | | | | |
|---|---|---|---|---|---|---|---|---|
| Metrics | Efficacy | | Generalization | | Specificity | | | |
| Methods | Success ↑ | Accuracy ↑ | Success ↑ | Accuracy ↑ | $D_{KL}$ ↓ | Top-1 ↑ | Top-5 ↑ | Top-10 ↑ |
| OneLayer | 74.4 | 43.3 | 58.7 | 19.6 | 0.55 | 68.8 | 65.7 | 64.4 |
| MEMIT | 94.0 | 79.7 | 82.1 | 45.3 | 0.81 | 63.5 | 60.8 | 59.4 |
| BLUE | 94.3 | 82.6 | 82.7 | 47.2 | 0.75 | 67.7 | 62.4 | 60.1 |
| WISE | 24.6 | 0.4 | 26.6 | 0.8 | **0.21** | **80.5** | **74.9** | **74.0** |
| PRUNE | 93.8 | 79.6 | 82.2 | 45.3 | 0.80 | 63.5 | 60.9 | 59.5 |
| RECT | 93.4 | 78.9 | 81.5 | 44.6 | 0.80 | 63.7 | 61.1 | 59.7 |
| Alphaedit | 99.4 | 97.2 | 92.1 | 58.7 | 0.98 | 62.9 | 57.8 | 55.9 |
| FE (ours) | **99.6** | **98.4** | **93.6** | **60.2** | 0.71 | 66.9 | 63.2 | 61.7 |

| | The ZsRE dataset | | | | | | | |
|---|---|---|---|---|---|---|---|---|
| Metrics | Efficacy | | Generalization | | Specificity | | | |
| Methods | Success ↑ | Accuracy ↑ | Success ↑ | Accuracy ↑ | $D_{KL}$ ↓ | Top-1 ↑ | Top-5 ↑ | Top-10 ↑ |
| OneLayer | 66.5 | 45.8 | 60.7 | 39.6 | 0.94 | 69.0 | 44.4 | 47.0 |
| MEMIT | 81.4 | 65.8 | 74.4 | 56.7 | 0.90 | 70.6 | 47.1 | 49.5 |
| BLUE | 82.5 | 67.8 | 76.8 | 60.1 | 0.81 | 77.4 | 46.4 | 48.5 |
| WISE | 9.1 | 0 | 9.2 | 0 | 2.48 | 8.0 | 4.8 | 10.5 |
| PRUNE | 81.4 | 65.7 | 74.3 | 56.5 | 0.90 | 70.5 | 47.2 | 49.6 |
| RECT | 80.7 | 64.6 | 73.5 | 55.4 | 0.89 | 70.7 | 47.5 | 49.8 |
| Alphaedit | 92.0 | 81.8 | 83.2 | 67.4 | 0.93 | 73.0 | 45.0 | 48.9 |
| FE (ours) | **93.7** | **84.6** | **85.1** | **69.8** | **0.47** | **91.5** | **58.2** | **60.0** |

In the following steps, we will proof that

$$\delta_{m_l}^\top J_{l \to L} \delta_{m_l} = \delta_{m_l}^\top S \delta_{m_l} \tag{34}$$

We have

$$\delta_{m_l}^\top J_{l \to L} \delta_{m_l} = \delta_{m_l}^\top (S + A) \delta_{m_l} = \delta_{m_l}^\top S \delta_{m_l} + \delta_{m_l}^\top A \delta_{m_l}, \tag{35}$$

So, Eq. 34 is equal to $\delta_{m_l}^\top A \delta_{m_l} = 0$. Since we have (note that $\delta_{m_l}^\top A \delta_{m_l}$ is a scalar)

$$\delta_{m_l}^\top A \delta_{m_l} = (\delta_{m_l}^\top A \delta_{m_l})^\top = \delta_{m_l}^\top A^\top \delta_{m_l} = \delta_{m_l}^\top (-A) \delta_{m_l} = -\delta_{m_l}^\top A \delta_{m_l}, \tag{36}$$

we know $2\delta_{m_l}^\top A \delta_{m_l} = 0$. Thus Eq. 34 holds. As a result, we have

$$\delta_{m_l}^\top J_{l \to L} \delta_{m_l} > 0 \iff \delta_{m_l}^\top S \delta_{m_l} > 0. \tag{37}$$

We know that (the definition of positive definite for a symmetric matrix)

$$S > 0 \iff \forall \delta_{m_l} \neq 0, \ \delta_{m_l}^\top S \delta_{m_l} > 0. \tag{38}$$

As a result, we know that

$$S > 0 \implies \cos(\delta_{m_L}^l, \delta_{m_l}) > 0. \tag{39}$$

The proof is completed.

### A.8. Results on GPT2-XL and GPT-J

The results with the GPT2-XL model and GPT-J model are in Table 6 ~ 7. We do not report results for RLEdit on these two models, as its performance is highly sensitive to hyperparameter choices. Despite extensive tuning efforts, we were unable to identify stable and appropriate hyperparameters. To avoid potential confusion arising from suboptimal configurations and unfair performance comparisons, we therefore omit RLEdit results for these models.

*Table 7.* Effectiveness of our FE. The model is GPT-J-6B.

| | The MCF dataset | | | | | | | |
| --- | --- | --- | --- | --- | --- | --- | --- | --- |
| Metrics | Efficacy | | Generalization | | Specificity | | | |
| Methods | Success ↑ | Accuracy ↑ | Success ↑ | Accuracy ↑ | $D_{KL}$ ↓ | Top-1 ↑ | Top-5 ↑ | Top-10 ↑ |
| OneLayer | 96.6 | 85.4 | 79.1 | 41.0 | 0.42 | 73.4 | 69.1 | 68.7 |
| MEMIT | 99.8 | 99.2 | 91.8 | 61.9 | 0.44 | 75.0 | 71.2 | 70.8 |
| BLUE | 99.8 | 99.1 | 94.9 | **69.2** | **0.38** | 78.0 | 72.9 | 72.4 |
| WISE | 27.3 | 6.8 | 50.9 | 15.2 | 4.47 | 45.8 | 48.0 | 49.5 |
| PRUNE | 99.8 | 99.3 | 91.8 | 61.8 | 0.43 | 75.1 | 71.2 | 70.9 |
| RECT | 99.8 | 99.2 | 91.5 | 61.2 | 0.43 | 75.1 | 71.3 | 71.0 |
| Alphaedit | 99.8 | 98.6 | 93.2 | 61.5 | 0.58 | 72.5 | 66.0 | 65.1 |
| FE (ours) | **99.9** | **99.4** | **95.4** | 65.7 | 0.42 | **78.4** | **73.6** | **72.8** |

| | The ZsRE dataset | | | | | | | |
| --- | --- | --- | --- | --- | --- | --- | --- | --- |
| Metrics | Efficacy | | Generalization | | Specificity | | | |
| Methods | Success ↑ | Accuracy ↑ | Success ↑ | Accuracy ↑ | $D_{KL}$ ↓ | Top-1 ↑ | Top-5 ↑ | Top-10 ↑ |
| OneLayer | 93.8 | 86.8 | 86.9 | 75.6 | 0.48 | 96.5 | 52.9 | 57.8 |
| MEMIT | 99.3 | 98.1 | 95.3 | 90.1 | 0.23 | 99.3 | 63.6 | 68.9 |
| BLUE | 99.3 | 98.2 | **96.4** | **92.6** | 0.16 | 100.0 | 68.6 | 74.2 |
| WISE | 51.8 | 21.7 | 49.0 | 19.0 | 10.87 | 2.2 | 2.5 | 2.6 |
| PRUNE | 99.4 | 98.2 | 95.1 | 90.0 | 0.24 | 99.2 | 63.7 | 68.9 |
| RECT | 99.2 | 97.9 | 95.1 | 89.8 | 0.24 | 99.1 | 63.9 | 68.9 |
| Alphaedit | 88.2 | 75.0 | 76.2 | 57.6 | 0.49 | 97.9 | 54.6 | 59.4 |
| FE (ours) | **99.5** | **98.8** | 96.1 | 92.2 | **0.08** | **100.0** | **74.0** | **78.6** |

