# OpenReview forum: "From Backward Spreading to Forward Replay: Revisiting Target Construction in LLM Parameter Editing"
_ICML.cc/2026/Conference — ICML 2026 regular_

### Official Review · Reviewer_jNQV · 2026-02-24

**Soundness:** 3
**Presentation:** 3
**Significance:** 3
**Originality:** 3
**Overall Recommendation:** 4
**Confidence:** 4

**Summary:**

The paper revisits target construction in locate-then-edit model editing. It argues that backward residual spreading is directionally inconsistent, and proposes Forward Replay (FE): optimize an early-layer anchor and propagate it forward to get layer-wise targets (Section 4 and Section 5). This is highly relevant to the ROME/MEMIT family.[1][2]

**Compliance With Llm Reviewing Policy:**

Affirmed.

**Key Questions For Authors:**

1. In Section 6.2, can you add stronger sequential-edit and multi-hop benchmarks, not only single-round settings?
2. Can you isolate FE behavior in high-conflict edits (large fact distance, contradictory edit chains)?
3. Can you sharpen the novelty statement relative to BLUE in Section 2/5 with explicit method-level differences?[3]
4. Can you include stricter locality evaluation protocols in line with recent analysis work?[6]

## References
[1] Meng et al. *Locating and Editing Factual Associations in GPT (ROME)*. arXiv:2202.05262. https://arxiv.org/abs/2202.05262

[2] Meng et al. *Mass-Editing Memory in a Transformer (MEMIT)*. arXiv:2210.07229. https://arxiv.org/abs/2210.07229

[3] Li et al. *Rethinking Residual Distribution in Locate-then-Edit Model Editing*. arXiv:2502.03748. https://arxiv.org/abs/2502.03748

[4] Fang et al. *AlphaEdit: Null-Space Constrained Knowledge Editing for Language Models*. arXiv:2410.02355. https://arxiv.org/abs/2410.02355

[5] Wang et al. *WISE: Rethinking the Knowledge Memory for Lifelong Model Editing of Large Language Models*. arXiv:2405.14768. https://arxiv.org/abs/2405.14768

[6] Liu et al. *Are We Evaluating the Edit Locality of LLM Model Editing Properly?* arXiv:2601.17343. https://arxiv.org/abs/2601.17343

[7] Mitchell et al. *Fast Model Editing at Scale (MEND)*. arXiv:2110.11309. https://arxiv.org/abs/2110.11309

[8] Mitchell et al. *Memory-Based Model Editing at Scale (SERAC)*. arXiv:2206.06520. https://arxiv.org/abs/2206.06520

**Limitations:**

`Yes.` Main limitations are acknowledged, but sequential-edit robustness and failure cases should be expanded.

**Strengths And Weaknesses:**

### Strengths
1. The paper focuses on a core mechanism in LTE editing (target construction), not a minor engineering tweak (Section 4/5).[1][2]
2. FE is simple to integrate into existing editing pipelines and does not require redesigning the full framework.
3. Section 6 reports consistent directional gains across efficacy/generalization/specificity metrics.

### Weaknesses
1. The theoretical analysis in Section 4.1 is mainly local/first-order. For strongly nonlinear regions, this may overestimate how stable the forward replay assumption is.
2. Section 5 places most control on the first edited layer. In sequential editing settings, this may increase early-layer interference risk, but Section 6.2 does not fully stress-test that regime.[4][5]
3. On novelty framing, Section 2 and Section 5 need a cleaner boundary versus BLUE, which already critiques residual-distribution issues.[3]
4. Section 6.2 still relies on conventional locality/specificity reporting. Recent evidence suggests some standard edit-locality metrics can be misleading.[6]

---

> ### Author Rebuttal · Authors · 2026-03-31
>
> Thank you very much for your thoughtful and constructive feedback.
>
> **W1 (local/first-order assumption)**. We think this is a misunderstanding. The local/first-order assumption is irrelevant to our FE, but only used to better understand the naive backward spreading: The eigenvector condition in Theorem 1 makes explicit why naive backward spreading is a strong assumption, and the positive-definite symmetric-Jacobian condition in Theorem 2 gives a plausible explanation for why the heuristic may still work locally in practice.
>
> The effectiveness of FE is very direct to show, which is why we omitted the proof.
>
> Suppose the input is $X = [x_1, x_2, ..., x_d, ..., x_n]$, where $x_d$ is the decisive token, with label Y, and let a three-layer neural network be defined as $f_3(f_2(f_1(X))) = \hat{Y}$. By gradient descent, we find an optimal $x_d = m_1$ such that $f_3(f_2(f_1(x_1, x_2, ..., m_1, ..., x_n))) = Y^\*$,
> where $Y^\*$ denotes a desirable output that yields a small cross-entropy $H_c(Y, Y^\*)$.
>
> Denote $H_1 = f_1(x_1, x_2, ..., m_1, ..., x_n) = [h_1, h_2, ..., h_d, ..., h_n]$. Then it follows immediately that
> $f_3(f_2(h_1, h_2, ..., h_d, ..., h_n)) = Y^\*$,
> simply by the basic property of functions: the same input cannot produce two different outputs.
>
> If we then take $h_d$ and define it as $m_2$, i.e., $m_2=h_d$, it follows directly that
> $f_3(f_2(h_1, h_2, ..., m_2, ..., h_n)) = Y^\*$.
> Therefore, once the edit target $m_1$ for the first layer is known, we can directly and exactly obtain the edit target $m_2 = h_d$ for the second layer through a strictly accurate forward computation. This is the theoretical basis of FE.
>
> Since this argument is quite direct, we did not elaborate on it in detail. Nevertheless, Appendix A.3 already conveys this intuition, and we will add the above discussion in the revision.
>
> **W2&Q1 (control on the first edited layer, and sequential editing)**.
>
> Thank you for this question. We may not have fully understood your point, so please let us clarify our interpretation.
>
> If by “most control on the first edited layer” you mean that we use the first decisive layer as the anchor point and solve for it by gradient descent, then yes, that is exactly our design. However, as explained in our response to W1 above, using the first decisive layer as the anchor point does not reduce the accuracy of the edit targets for later layers. Through FE, the targets for subsequent layers are obtained by exact forward computation. In contrast, existing methods effectively place the anchor point at the last decisive layer and then propagate backward, which is less accurate.
>
> Regarding sequential editing, the main challenge is that later edits may interfere with knowledge edited earlier. This falls under locality, so its central challenge is to protect other knowledge (the second term in Equation 4). In contrast, the only difference between our method and MEMIT lies in how to better obtain m in the first term of Eq. 4, that is, how to more accurately inject the target knowledge into the model. These are two orthogonal aspects.
>
> As for how our method could be extended to sequential editing, sequential editing methods usually involve additional mechanisms specifically designed to protect previously edited knowledge. For example, AlphaEdit maintains an extra historical knowledge matrix, and HSE [1] maintains a Fisher information matrix to preserve past edits. Our method is very simple and does not conflict with such additional protection mechanisms, so it can potentially be combined with them to adapt to sequential editing. However, since our main focus is not on protecting other knowledge, and our method does not introduce additional mechanism that would cause more damage to other knowledge compared with standard MEMIT (in fact, **we get improved locality scores**), this direction is currently beyond the scope of this paper, and we leave it for future work.
>
> **W3&Q3 (boundary versus BLUE)**. Thank you for the valuable suggestion. In L71–91, we specifically discuss three limitations of BLUE, which lead to its doubled time complexity (please see our response to W1 of Reviewer #1ueMw) and unstable performance (please see our response to Q5 of Reviewer #2HWeF).
>
>
> **W4&Q4 (evaluation protocols in line with [6])**. Thank you for the valuable suggestion, but we think this is a misunderstanding. In fact, the metrics we use are exactly those proposed in [6], as stated in L357–365 (right column).
>
>
> **Q2 (FE behavior in high-conflict edits)**. This is a valuable suggestion, but our experiments already included such high-conflict edits. The MCF dataset we use is a counterfactual dataset, in which the edit target is inherently in conflict with the old knowledge.
>
> [1] Hippocampal-like Sequential Editing for Continual Knowledge Updates in Large Language Models

---

> > ### Author Rebuttal · Reviewer_jNQV · 2026-04-05
> >
> > The authors provided a good response. I had already given a positive score previously. Considering the overall novelty and my initial comments, I believe the paper is worth accepting, and my current score is already sufficient.

---

### Official Review · Reviewer_hx2D · 2026-03-08

**Soundness:** 3
**Presentation:** 3
**Significance:** 2
**Originality:** 3
**Overall Recommendation:** 4
**Confidence:** 3

**Summary:**

This paper proposes Forward Replay (FE), a novel target construction strategy for Large Language Model parameter editing that addresses the theoretical inconsistency of the standard "backward spreading" approach. Instead of optimizing the final layer and propagating residuals backward, FE optimizes the target hidden states of the first editable layer and derives subsequent targets via standard forward propagation, ensuring intrinsic consistency across all edited layers without added computational cost. Experiments show that FE achieves state-of-the-art performance across multiple models and datasets, significantly outperforming existing baselines in editing efficacy, generalization, and specificity while demonstrating superior convergence behavior.

**Compliance With Llm Reviewing Policy:**

Affirmed.

**Final Justification:**

The authors have addressed my concerns. After re-examining the paper, I believe it is worthy of acceptance. Therefore, I will raise my score.

**Key Questions For Authors:**

1. My primary concern lies in the accuracy of editing the first layer. The backward propagation actually minimizes deviations across all edited layers. In contrast, by starting the editing process directly from the first layer, the proposed approach could amplify this error instead. Are there any experimental results that address this concern?

2. The experiments are conducted solely on the Llama3-8B-Instruct model. It remains unclear whether the proposed method generalizes to other architectures, such as Qwen, or to models of different sizes, which would help validate its broader applicability. While the results in Table 3 and Table 4 appear promising, further evaluation across a wider range of models would provide more compelling evidence.

3. Section 4.1 Theoretical grounding lacks a clear connection to the main proposal. It currently offers no theoretical evidence explaining why Forward Replay is superior to Backward Spreading. Could the authors expand this section to explicitly demonstrate the theoretical advantages?

**Limitations:**

yes

**Strengths And Weaknesses:**

**Strengths:**

1. The paper provides a compelling theoretical critique of the widely adopted backward spreading mechanism, identifying a fundamental inconsistency in how target states are constructed for multi-layer editing.

2. The proposed Forward Replay method is conceptually simple and elegant, ensuring intrinsic consistency across edited layers via standard forward propagation without adding computational overhead.

3. Experiments demonstrate that the method achieves state-of-the-art performance across multiple models and datasets, significantly outperforming existing baselines in editing efficacy and generalization.


**Weaknesses:**

1. The proposed FE is essentially a minor modification of MEMIT. It modifies each layer sequentially through forward propagation, which heavily relies on the accuracy of the edit applied to the first layer. If the authors argue that backward spreading leads to large directional deviations and poor convergence, then the deviation in the first-layer edit in FE might be even more pronounced, making it harder to guarantee correctness. Are there any experiments that address this issue?

2. The experimental evaluation is limited to only two datasets (MCF and ZsRE), which restricts the assessment of the method's generalizability to more diverse or complex editing scenarios.

3. Ablation studies are insufficient to isolate the specific contribution of the forward propagation mechanism. Are there ablation results of optimizing only a few subsequent layers, that reduce computational demands as other works?

---

> ### Author Rebuttal · Authors · 2026-03-31
>
> Thank you very much for your valuable comments and questions.
>
> **W1&Q1&Q3 (why does our method work better)**. We are very sorry for causing such a huge misunderstanding.
>
> As shown in Fig. 1, LTE consists of three stages. First, an anchor-point edit target m is obtained through gradient descent. Second, based on this anchor point, the edit targets (which can be understood as analogous to the target label in other NLP tasks) for the other layers are derived through a propagation procedure. Only in the third stage are the model parameters actually updated according to these edit targets. Regardless of how the first two stages are implemented, all methods, including FE, perform the third stage in the same way: they sequentially edit the model from the first decisive layer and proceed forward. This stage is shared by all methods. **Our method lies mainly in obtaining more accurate edit targets before the third stage**. In particular, FE can ensure that the propagation from the initial anchor-point edit target to the edit targets of subsequent layers is exact.
>
> Suppose the input is $X = [x_1, x_2, ..., x_d, ..., x_n]$, where $x_d$ is the decisive token, with label Y, and let a three-layer network be defined as $f_3(f_2(f_1(X))) = \hat{Y}$. By gradient descent, we find an optimal $x_d = m_1$ such that $f_3(f_2(f_1(x_1, x_2, ..., m_1, ..., x_n))) = Y^\*$,
> where $Y^\*$ denotes a desirable output that yields a small cross-entropy $H_c(Y, Y^\*)$.
>
> Denote $H_1 = f_1(x_1, x_2, ..., m_1, ..., x_n) = [h_1, h_2, ..., h_d, ..., h_n]$. Then it follows immediately that
> $f_3(f_2(h_1, h_2, ..., h_d, ..., h_n)) = Y^\*$,
> simply by the basic property of functions: the same input cannot produce two different outputs.
>
> If we then take $h_d$ and define it as $m_2$, i.e., $m_2=h_d$, it follows directly that
> $f_3(f_2(h_1, h_2, ..., m_2, ..., h_n)) = Y^\*$.
> Therefore, once the edit target $m_1$ for the first layer is known, we can directly and exactly obtain the edit target $m_2 = h_d$ for the second layer through a strictly accurate forward computation. This is the theoretical basis of FE.
>
> This is straightforward, thus we did not elaborate on it in detail. Nevertheless, Appendix A.3 already conveys this intuition to some extent and we will add the above discussion in our revision.
>
> **W2&Q2 (models and datasets)**. We are sorry to cause such a misunderstanding. In fact, we evaluate on three models spanning three different scales—8B, 6B, and 1.5B: Llama3-8B, GPT-J, and GPT2-XL (L323–328, due to space limitations, the experiments on the latter two models are presented in Appendix A.7, but the improvements on GPT2 are even more significant). This is consistent with the baselines AlphaEdit and BLUE. The choice of the two datasets, MCF and ZsRE, also follows standard practice in this line of work, as adopted by both AlphaEdit and BLUE.
>
> Following your suggestion, we further add two more datasets: WikiCounterFact (WCF) and MQuAKE. The results with Llama 3 are shown below. These two datasets do not provide a test set for locality evaluation.
> These results further support the effectiveness of FE beyond MCF and ZsRE.
>
> |WCF|OneLayer|MEMIT|BLUE|WISE|PRUNE|RECT|ALphaedit|FE|
> |-|-|-|-|-|-|-|-|-|
> |Eff-Succ|80.4|84.8|88.6|33.4|84.9|83.5|81.2|**94.0**|
> |Eff-Acc|38.8|57.7|65.8|0.6|57.8|55.0|46.2|**85.0**|
> |Gen-Succ|48.8|56.8|71.4|47.0|56.6|55.0|53.2|**79.3**|
> |Gen-Acc|5.9|11.9|25.9|0.6|12.2|11.4|9.5|**40.6**|
>
> |MQuAKE|OneLayer|MEMIT|BLUE|WISE|PRUNE|RECT|ALphaedit|FE|
> |-|-|-|-|-|-|-|-|-|
> |Eff-Succ|88.6|93.4|96.8|16.8|93.4|93.0|90.3|**99.0**|
> |Eff-Acc|77.4|89.6|90.8|9.0|89.8|88.6|82.4|**98.0**|
> |Gen-Succ|76.4|80.0|84.8|8.1|80.1|79.6|77.0|**86.8**|
> |Gen-Acc|52.0|63.0|69.7|1.5|63.0|61.9|57.0|**72.2**|
>
>
>
> **W3 (Ablation studies are insufficient to isolate the specific contribution of the forward propagation mechanism. Are there ablation results of optimizing only a few subsequent layers, that reduce computational demands as other works?)**. We are sorry, but we do not fully understand your point, so please correct us if we are interpreting it incorrectly. In model editing, the optimization in Equation 4 consists of two parts: first, obtaining a good edit target m (which can be viewed as analogous to a target label in a classification task); and second, solving Equation 4 once m is obtained. The only difference between FE and MEMIT is how this target m is derived; all other parts of the optimization are the same. In this sense, MEMIT already serves as the direct ablation baseline for FE.
>
> Moreover, the computational complexity of FE is almost the same as that of MEMIT, as shown in our response to W1 of Reviewer #1 (ueMw). For this reason, we are not fully sure what is meant here by “optimizing only a few subsequent layers” to reduce computational cost. Could you explain more specifically what you mean by “optimizing only a few subsequent layers, that reduce computational demands as other works,” so that we can better understand and address your concern?

---

> > ### Author Rebuttal · Reviewer_hx2D · 2026-04-03
> >
> > The authors have addressed my concerns. However, based on the overall innovation and feedback of the paper, I tend to maintain the original score.

---

### Official Review · Reviewer_HWeF · 2026-03-12

**Soundness:** 4
**Presentation:** 2
**Significance:** 3
**Originality:** 3
**Overall Recommendation:** 5
**Confidence:** 4

**Summary:**

This paper studies target construction in LLM parameter editing methods. It analyzes potential issues of the commonly used backward spreading strategy and proposes a Forward Replay method that edits earlier layers and propagates the hidden states forward. The method can be easily integrated with existing frameworks and shows improved empirical performance.

**Compliance With Llm Reviewing Policy:**

Affirmed.

**Final Justification:**

Thanks for the response. I keep the score of 5.

**Key Questions For Authors:**

1. In Theorem 2, it would be clearer to explicitly use symmetrization of the matrix, rather than symmetrix part.

2. Does Forward Replay introduce any potential stability issues when applied to very deep models?

3. Typo in line 752 (“prove” should be “proof”).

4. Section 4 contains two paragraphs titled “Empirical Verification”.

5. BLUE is a strong baseline, whose results is very close to the proposed method. Have you tried BLUE+FE? Can this combination give better results?

**Limitations:**

Yes

**Strengths And Weaknesses:**

Strengths:

1. The paper provides a useful analysis explaining why backward spreading may not always yield optimal editing performance.

2. The proposed Forward Replay approach is conceptually simple and easy to integrate with existing editing frameworks.

3. Experiments show that the proposed method can significantly improve performance over existing editing methods while maintaining simplicity as an add-on component.

Weakness:

1. The paper suffers from poor writing and organization. The structure and explanation of the proposed method could be improved, as the main idea is difficult to understand on first reading, even though the proposed idea is simple and intuitive.
2. Better to have a slightly more theoretical explanation of why the proposed method works better. For instance, analyze a simple 2-layer case to show the forward-replay can do better editing.

---

> ### Author Rebuttal · Authors · 2026-03-31
>
> Thank you very much for your careful review and helpful suggestions.
>
> **W1 (structure)**. Owing to space constraints, we placed the detailed explanation in Appendix A.3 (as noted in L327). In the revision, we will reorganize the method section so that a broader audience, including readers unfamiliar with model editing, can more easily follow the paper.
>
> **W2 (theoretical explanation)**.
> The effectiveness of FE is straightforward, which is why we omitted a formal proof.
>
> Suppose the input is $X = [x_1, x_2, ..., x_d, ..., x_n]$, where $x_d$ is the decisive token, with label Y, and let a three-layer neural network be defined as $f_3(f_2(f_1(X))) = \hat{Y}$. By gradient descent, we find an optimal $x_d = m_1$ such that $f_3(f_2(f_1(x_1, x_2, ..., m_1, ..., x_n))) = Y^\*$,
> where $Y^\*$ denotes a desirable output that yields a small cross-entropy $H_c(Y, Y^\*)$.
>
> Denote $H_1 = f_1(x_1, x_2, ..., m_1, ..., x_n) = [h_1, h_2, ..., h_d, ..., h_n]$. Then it follows immediately that
> $f_3(f_2(h_1, h_2, ..., h_d, ..., h_n)) = Y^\*$,
> simply by the basic property of functions: the same input cannot produce two different outputs.
>
> If we then take $h_d$ and define it as $m_2$, i.e., $m_2=h_d$, it follows directly that
> $f_3(f_2(h_1, h_2, ..., m_2, ..., h_n)) = Y^\*$.
> Therefore, once the edit target $m_1$ for the first layer is known, we can directly and exactly obtain the edit target $m_2 = h_d$ for the second layer through a strictly accurate forward computation. This is the theoretical basis of FE.
>
> Since this argument is quite direct, we did not elaborate on it in detail. Nevertheless, Appendix A.3 already conveys this intuition, and we will add the above discussion in the revision.
>
> **Q2 (stability issues)**.  In principle, we do not believe forward replay introduces stability issues compared with other LTE methods. All LTE methods require solving for an initial anchor point m by gradient descent. Our only difference is that we place this anchor at the first decisive layer rather than the last. Since the number of decisive layers is usually small relative to the total number of layers in an LLM (e.g., 5 out of 32 in Llama3-8B), this difference is minimal.
>
> The subsequent forward replay is highly accurate, as discussed in W2 above. Forward propagation is also generally very stable, which is why low-precision quantization is often sufficient during inference. In contrast, other LTE methods rely on backward spreading from the anchor point, which becomes less accurate as the propagation distance increases, as verified in Tables 1–2. We do not have GPUs to edit larger models.
>
> **Q5 (BLUE baseline)**. Our method cannot be combined with BLUE, because it is also a way of constructing edit targets. Our method performs consistently well, whereas BLUE is strong in some settings but weak in others. Under the MCF + Llama3 setting, BLUE achieves results similar to ours, possibly because the editing task in that setting is relatively easy, leaving limited room for improvement since most methods already perform strongly. However, on ZsRE, especially under the more challenging accuracy metrics, our gains over BLUE are substantial. In addition, in the GPT2-XL experiments reported in Table 6 (L771–794), we outperform BLUE by more than 10% on many metrics.
>
> We scale the MCF + Llama3 setting to 10,000 examples. Under this more challenging setting, the advantage of our method becomes even more significant:
> ||Eff-Succ|Eff-Acc|Gen-Succ|Gen-Acc|KL ↓|Top1|Top5|Top10|
> |-|-|-|-|-|-|-|-|-|
> |BLUE|96.2|89.4|85.1|53.2|1.18|60.4|57.5|55.9|
> |FE|99.2|97.7|91.2|58.6|1.09|64.5|61.2|59.9|
>
> Both FE and MEMIT first use gradient descent to find an anchor point m and then propagate it to other layers. BLUE, by contrast, independently runs gradient descent for each layer to obtain that layer’s edit target. In practice, due to computational cost, it does this only for the first and last decisive layers. Even so, its runtime is still roughly doubled.
> || Time (hour) |
> |-|-|
> | MEMIT | 0.61|
> | BLUE  | 1.16|
> | FE| 0.63 |
>
>
> More importantly, m is high-dimensional and the solution found by gradient descent is not unique, thus edit targets computed independently for different layers are not necessarily mutually compatible. Intuitively, suppose the desired output is Y*. The edit target at each layer can be viewed as lying on a circular orbit around Y*, and any point on that orbit could serve as the edit target for that layer. In FE, we first backpropagate Y* to the outermost orbit to find one point m, and then use the corresponding propagation path to determine the intersection points on the other orbits as the edit targets for the remaining layers. In this way, the targets are guaranteed to be mutually compatible. BLUE, however, independently backpropagates Y* to each orbit, which does not guarantee that the propagation directions are consistent across layers.
>
> **Q1,3,4 (typos)**. Thank you for the careful review. We will correct them in the revision.

---

> > ### Author Rebuttal · Reviewer_HWeF · 2026-04-03
> >
> > The authors have answered the raised questions well.

---

### Official Review · Reviewer_ueMw · 2026-03-13

**Soundness:** 3
**Presentation:** 2
**Significance:** 3
**Originality:** 3
**Overall Recommendation:** 4
**Confidence:** 3

**Summary:**

This paper studies target construction in locate-then-edit LLM parameter editing. It analyses the limitations of the widely used backward-spreading heuristic from a local Jacobian perspective, and proposes a simple alternative, FE, which instead optimises the anchor hidden state at the first decisive layer and uses forward replay to recover compatible targets for subsequent layers. Experiments on MCF and ZsRE show that FE improves MEMIT and also transfers to PRUNE, RECT, and AlphaEdit, with particularly strong results on LLaMA3-8B-Instruct and supporting appendix evidence on additional models.

**Compliance With Llm Reviewing Policy:**

Affirmed.

**Key Questions For Authors:**

1 The paper claims that FE has essentially the same computational cost as standard MEMIT, but this is argued rather than directly measured. Could the authors report wall-clock time, peak memory, and throughput comparisons against MEMIT and BLUE? If FE indeed has near-identical runtime and memory overhead, this would strengthen both my soundness and significance assessment.

2 The current experiments focus mainly on MCF and ZsRE under batch editing. Could the authors evaluate FE in more realistic or broader settings, such as open-domain or context-rich evaluation, to test whether the gains persist beyond the current benchmark regime? A positive result here would materially increase my confidence in the method’s practical significance.

3 Have the authors tested FE under long-horizon sequential editing, or do they have a principled reason to expect improved stability in that setting? Since target compatibility across layers is a central claim, evidence of better behaviour under many sequential edits would substantially strengthen the paper.

4 The theoretical analysis explains why backward spreading may fail, but it is less clear when first-layer anchoring and forward replay may themselves become unreliable. Could the authors clarify the main failure cases of FE, and whether there are settings in which it underperforms BLUE or standard backward spreading? A clearer discussion here would improve my assessment of soundness.

**Limitations:**

No. The limitations discussion is too brief, and the impact statement is effectively empty.

**Strengths And Weaknesses:**

This paper isolates a genuinely important and previously under-analysed component of LTE editing: the construction of multi-layer targets rather than merely the final closed-form weight update. I think it is a meaningful technical question. The eigenvector condition in Theorem 1 makes explicit why naive backward spreading is a strong assumption, and the positive-definite symmetric-Jacobian condition in Theorem 2 gives a plausible explanation for why the heuristic may still work locally in practice. Overall, the method is conceptually clean. FE changes only the propagation scheme for layer-wise targets, leaving the rest of the LTE pipeline intact. That makes the proposal easy to understand and, at least in principle, easy to retrofit into existing methods. Besides, the empirical diagnostics on backward spreading are helpful. The cosine-similarity analysis and residual-ratio experiments give concrete evidence that distortion worsens with distance from the anchor layer and that early backward-spread layers can contribute little or even harm progress. The experimental results are strong. The paper’s evaluation protocol is somewhat more modern than previous work. It reports both teacher-forced “Success” and free-generation “Accuracy”, and it adopts behaviour-deviation specificity metrics rather than relying only on conventional ground-truth locality scores. That is directionally aligned with recent criticism of legacy evaluation practice. Also, the claimed computational overhead is plausibly low, because the extra work is only a short forward replay across a small number of decisive layers rather than repeated per-layer optimisation, and it is attractive.

However, the theoretical analysis explains why backward spreading can fail, but it does not rigorously establish that forward replay is optimal, or even near-optimal, under realistic non-linear model dynamics. The local Jacobian treatment is informative, yet still a first-order argument around small perturbations.  In addition, this paper argues that the extra forward pass is negligible, but it does not report wall-clock time, memory, or throughput comparisons against MEMIT and BLUE. For a systems-facing editing paper, that omission matters. Besides, I suggest test long-horizon sequential editing. That is a significant gap because recent work shows abrupt collapse in LTE methods under many sequential edits due to norm growth and instability. Also, FE is architecturally designed as an LTE plug-in, so these baselines mainly show that unrelated paradigms behave differently under the selected setup, rather than offering a strict apples-to-apples test of the proposed idea.

---

> ### Author Rebuttal · Authors · 2026-03-31
>
> Thank you very much for your careful review and constructive comments.
>
> **W1 (computational cost)**.
> |       | Time (hour) | Peak Memory (GB) |
> |-------|------------|------------------|
> | MEMIT | 0.61       | 16.86            |
> | BLUE  | 1.16       | 16.91            |
> | FE    | 0.63       | 16.95            |
>
> The above table compares the time and peak memory required by different methods to compute the edit targets (i.e., $m_i$ in Eq. 4) for 2,000 data samples  using Llama 3-8B.
> As shown, compared with MEMIT, our method introduces only a very small additional cost: about 0.02 hours in runtime (3%) and 0.09 GB in peak memory (0.5%). In contrast, BLUE requires nearly twice as much time as MEMIT.
>
>
>
> **W2 (context-rich evaluation)**.
> | Efficacy | OneLayer | MEMIT | BLUE | WISE | RLEdit | PRUNE | RECT | Alphaedit | FE   |
> |----------|----------|-------|------|------|--------|-------|------|-----------|------|
> | Success  | 79.5     | 87.5  | 88.2 | 53.8 | 67.5   | 87.3  | 86.8 | 85.6      | **92.2** |
> | Accuracy | 60.4     | 73.4  | 75.2 | 15.3 | 41.6   | 76.2  | 72.3 | 70.1      | **82.6** |
>
> | Generalization | OneLayer | MEMIT | BLUE | WISE | RLEdit | PRUNE | RECT | Alphaedit | FE   |
> |----------------|----------|-------|------|------|--------|-------|------|-----------|------|
> | Success        | 75.7     | 84.2  | 86.1 | 52.6 | 66.2   | 84.1  | 83.6 | 82.4      | **89.6** |
> | Accuracy       | 54.8     | 67.6  | 70.9 | 14.8 | 39.6   | 67.4  | 66.6 | 64.7      | **77.4** |
>
> Above are the results of the context-guided evaluation conducted following Appendix A.7 of [1]. The dataset and model are ZsRE and Llama3, respectively, corresponding to the results in our Table 3 (the context-guided evaluation in [1] was specifically designed for QA datasets and is therefore not suitable for MCF) . As shown, our method still maintains a clear advantage under this setting.
>
> **W3 (long-horizon sequential editing)**. The main issue in sequential editing is that later edits may disrupt knowledge that has already been edited earlier. This falls under locality, so its central challenge is to protect other knowledge (the second term in Equation 4). In contrast, our work focuses on how to better obtain $m_i$ in the first term of Equation 4, that is, how to correctly inject the current knowledge into the model. These are two orthogonal aspects.
>
> As for extending our method to sequential editing, such settings usually require additional mechanisms specifically designed to preserve previously edited knowledge. For example, AlphaEdit maintains an extra historical knowledge matrix, and HSE [2] maintains a Fisher information matrix to preserve past edits. Our method is simple and does not conflict with such protection mechanisms, so it could potentially be combined with them for sequential editing.
>
> However, our main focus is not on preserving previously edited knowledge. Moreover, our method does not introduce any additional mechanism that would increase interference with other knowledge compared with standard MEMIT; in fact, we achieve improved locality scores. Therefore, this direction is beyond the scope of the current paper, and we leave it for future work.
>
> **W4 (whether there are settings in which it underperforms BLUE or standard backward spreading)**. The effectiveness of FE is very direct to see, which is why we omit a detailed proof. It is equivalent to the following: we have an input that passes through n black boxes in sequence and finally produces a loss $l$. By gradient descent, we find a good input $x$ such that, after passing through these n black boxes, we get a small final loss $l$. FE then simply picks out the intermediate states along this same computation path. Clearly, if these intermediate states are fed through the remaining black boxes, they will produce the same final loss $l$, which is therefore also very small.
>
> Thus, once a good anchor point at the first layer has been found, FE directly reuses the corresponding forward computation path, rather than approximating targets through backward spreading. For this reason, we would not expect FE to underperform standard backward spreading in principle.
>
> If you are interested in more detailed theoretical basis of FE, please refer to our response to W2 for reviewer #2 (HWeF).
>
> [1] The Mirage of Model Editing: Revisiting Evaluation in the WILD
> [2] Hippocampal-like Sequential Editing for Continual Knowledge Updates in Large Language Models

---

> > ### Author Rebuttal · Reviewer_ueMw · 2026-04-02
> >
> > Thanks to the authors for providing explanations. The overall impression for the paper remains positive.

---

### Decision · Program_Chairs · 2026-04-30

**Decision:**

Accept (regular)

**Comment:**

All reviewers gave positive scores and the authors' rebuttals solve the raised problems well.